# Thymosin Beta 15 Alters the Spatial Development of Thymic Epithelial Cells

**DOI:** 10.3390/cells11223679

**Published:** 2022-11-19

**Authors:** Xie Xu, Kai He, Robert D. Hoffman, Yuyuan Ying, Nana Tao, Wenqin Guo, Jiaman Shen, Xi Liu, Meiya Li, Meiqiu Yan, Guiyuan Lv, Jianli Gao

**Affiliations:** 1School of Pharmaceutical Sciences, Zhejiang Chinese Medical University, Hangzhou 310053, China; 2The First Affiliated Hospital, School of Medicine, Zhejiang University, Hangzhou 310009, China; 3Department of Traditional Chinese Medicine, Yo San University of Traditional Chinese Medicine, Los Angeles, CA 90066, USA; 4Department of Traditional Chinese Medicine, The Second Affiliated Hospital of Wenzhou Medical University, Wenzhou 325000, China; 5Academy of Chinese Medical Sciences, Zhejiang Chinese Medical University, Hangzhou 310053, China; 6State Key Laboratory of Quality Research in Chinese Medicine, University of Macau, Macao 999078, China

**Keywords:** action binding protein, reticular differentiation, thymocytes, thymosin beta 15, thymic epithelial cells

## Abstract

The thymus is the most sensitive organ under various pathophysiological conditions, such as aging, starvation, and infection. As a key stromal cell for T cell development, it is well-known that thymic epithelial cells (TECs) play an important role in the thymus response to the external environment. Thymosin beta 15 (Tβ15) is a G-actin binding protein secreted by TECs, it plays an important role in maintaining the dynamic balance of actin, angiogenesis, axonal formation, and wound healing, but the relationship between Tβ15 and TECs is not clear yet. Here, we show the impact of Tβ15 on the TEC’s spatial development, as well as the T-cell differentiation and thymic output. As a result, TEC is the main effector cell of Tβ15 in the thymus. Tβ15 OX inhibits the chemotaxis of TECs to the medulla and subsequently blocks the positive selection of thymocytes from CD3^+^TCRβ^+^CD4^+^CD8^+^ double positive cells to CD3^+^TCRβ^+^CD4^+^CD8^−^ single-positive (CD4SP) cells. Tβ15-knockdown accelerates the reticular differentiation of astral TECs and medullary TECs. Importantly, mice implanted with Tβ15-knockdown iTECs show high thymic output but low peripheral T cell maturity and activity. In a word, our results explain the role of Tβ15 on the differentiation and function of TECs and provide a new perspective for understanding the process of thymus development and degeneration.

## 1. Introduction

The thymus is a key organ linked to senescence and immunity, and the dynamic balance of T-cell subsets and the polymorphisms of T-cell receptors are the recognized indicators of immunosenescence [1,2,3]. We and others have shown that the loss of corticomedullary demarcation, medullary atrophy, and decreased epithelial cells exist in the thymus of various mice with acute and chronic thymus involution, including natural aging and D-galactose-induced acute thymus involution [4]. This research suggests that the main pathological changes in thymus involution are the corticomedullary disorder of the thymus, especially the deregulation of the distribution and function of thymic epithelial cells (TECs), which trigger the imbalance of thymocyte differentiation.

The thymus is mainly composed of hematopoietic thymocytes and TECs [5,6]. The reticular structure of the TECs is critical to supporting the normal structure of the thymus, and it is also the cradle of T cell development, differentiation, selection, and maturation [7]. The abnormality of TECs directly affects the immune status of the body. Based on their phenotype and localization, the TECs in mice can be divided into two types: cortical TECs (cTECs) and medullary TECs (mTECs). Among them, mTECs are also known as medullary reticulated epithelial cells, and cTECs can be further divided into two types: subcapsular epithelial cells (subcapsular TECs) and astroepithelial cells (aTECs). All TEC lineages derive from a common bipotent precursor of the endodermal origin that initially develops into cTEC-like phenotypes. During late ontogeny, the loss of cTEC-specific markers and potential leads to the generation of the medullary reticulated epithelium [8,9]. Subcapsular TECs and aTECs coordinate T cell development at early stages and positive selection. However, at the late stages of T cell development, medullary reticulated epithelial cells mainly perform negative selection, including deleting immature T cells that are highly reactive to tissue-limiting antigens (TRAs) [10,11,12]. 

As the main sequestering protein family of G-actin, the beta-thymosin family is an important peptide hormone secreted by the subcapsular TECs [13], which can play a key regulatory role in the immune system by paracrine means, especially in controlling the development and function of T lymphocytes [14,15]. Recent advances in thymus biology have indicated that the complex three-dimensional (3D) network of TECs involves the biological processes of muscle-like contractions, such as actin and myosin [16,17]. 3D stereoscopic TECs encapsulate various immature immune cells to constitute TEC compartments and are essential for thymus development, organization, and function. 

Thymosin beta 15 (Tβ15) is the thymosin with the highest affinity to actin and can effectively regulate the G-/F-actin ratio in cells. It is amorphous in the free state and can bind to the actin monomer in a 1:1 ratio. It ultimately affects actin remodeling and is closely related to cell motility and angiogenesis [18,19]. Tβ15 plays a key role in the disturbed thymic architecture and decreased output of mature T cells. However, the role of Tβ15 in the whole process of the motility and differentiation of TECs is still unknown. In the present work, we focus on the thymic compartment and the detailed phenotyping of thymocytes after the Tβ15 expression changes, and the result suggests that Tβ15 alters the development of TECs, causing subsequent changes in T cell development. Very interestingly, we find that Tβ15 interferes with the development of TCRβ^+^CD3^+^ CD4SP thymocytes and peripheral T lymphocyte homeostasis in the thymus by blocking the development of cTEC-like precursors into aTECs and mTECs. Our data provide a new direction not only for the study of thymic development but also for related studies in immunological senescence.

## 2. Materials and Methods

### 2.1. Animals and Cells

All the mice were purchased from SIPPR-BK Corporation (Shanghai, China), and all animal protocols were approved by the Institutional Animal Care and Use Committee at Zhejiang Chinese Medical University Laboratory Animal Research Center (Permit Number: SYXK (Zhejiang) 2021–0012) and were performed in accordance with the relevant institutional and national guidelines and regulations. Suckling C57BL/6 mice (female) were used to isolate thymocytes. C57BL/6 mice (female, 1–2 weeks old) were used for the thymus organ culture, and BALB/c nude mice (female, 4–6 weeks old) were used in organoid transplant experiments.

Immortalized mouse thymic epithelial cells were constructed according to our published protocol [20].

### 2.2. Cell Infection, Transfection, and Screening

For all adenovirus infections, polybrene (40 μg/mL, Sigma-Aldrich, St. Louis, MO, USA) was added to the culture medium to enhance the infection efficiency. For lentivirus infection, a selection of puromycin-resistant iTECs was carried out 48 h after transfection by the addition of 3 μg/mL puromycin (Sangon Biotechnology, Shanghai, China). Transfection efficiency was measured using RT-qPCR. All sequence information is provided in Appendix A.

### 2.3. In Vitro Mouse Thymus Organ Culture System

The thymus of the mice was extracted and isolated, then co-cultured with Tβ15 adenovirus (1 × 10^8^ vp/thymus) for 24 h at 37 °C. Tβ15 siRNA and NC siRNA (50 μM, GuanNan Biotechnology, Hangzhou, China) were transfected with 1.25 μL Lipo6000^TM^ transfection reagent (Beyotime, Shanghai, China). After being cultured in vitro for 7 days, all of the thymus organs were used in subsequent experiments.

### 2.4. Murine Artificial Thymic Organoid Cultures

Three-dimensional “cell-embedded” matrigel cultures were performed by creating a layer of cross-linked hyaluronic acid (HA, Hangzhou Jizhi Biotechnology Co., Ltd., Hangzhou, China) in 24-well plates (200 μL), which was then put back in a 37 °C incubator for 30 min to solidify. The primarily isolated mouse thymocytes and iTECs (20:1) were mixed, and a 2 × 10^5^ cells mixture in 100 μL medium was seeded in 24-well. Cells were allowed to attach for 30 min before adding an equal volume of the medium with DMEM/F12-B27 (Gibco, Northrend Biotechnology, Hangzhou, China) [21]. To assess the organoid forming efficiency (OFE), the number of organoids formed in each visual field was calculated, and their diameter was measured under an inverted microscope (Axio Observer. A1, Jena, Germany) on days 7, 14, and 21.

### 2.5. Subcutaneous Transplantation Experiments of Thymic Organoids

After two days of acclimatization, 100 μL thymic organoids were injected subcutaneously into the bilateral axilla of 4-week-old female BALB/c nu/nu mice. The ratio of thymocytes to iTECs was 20:1. The mice were sacrificed at weeks 1 and 2 after the implantation, respectively. The isolated organoids were fixed with 10% formalin for hematoxylin and eosin (H&E, Sigma-Aldrich, St. Louis, MO, USA) staining and immunohistochemical staining. A portion of the spleen was taken for *TRECs* quantification analysis, and the remaining spleen was used for the detection of peripheral T lymphocyte subsets.

### 2.6. Immunofluorescence Imaging

Formalin-fixed and paraffin-embedded tissues were sectioned at 4 μm. Tissue slices were placed in an EDTA antigen retrieval buffer. The processed sections were stained with primary antibodies as follows: mouse monoclonal Cytokeratin 8 (CK8; GR3181962-4, Abcam, Cambridge, UK; 1:200) antibody and the rat anti-mouse Cytokeratin 5 (CK5; 57u2005, Affinity, Changzhou, China; 1:200) antibody. The secondary antibodies were as follows: mouse anti-rabbit IgG-FITC (K2017, Santa Cruz, CA, USA; 1:200); F (ab’) 2-goat anti-mouse IgG (H+L) Alexa Fluor 594 (GAR48810900223, Thermo Fisher Scientific, Waltham, MA, USA; 1:200); and the blue fluorescence was a DAPI-labeled nucleus (Sigma-Aldrich, St. Louis, MO, USA). Pictures were taken with a digital pathological section scanner (OLYMPUS, VS120-S6-W, Tokyo, Japan).

### 2.7. Histology and Morphometric Analysis

The sections were stained with H&E. For immunohistochemical analysis, after deparaffinization and the antigen retrieval process (citrate antigen retrieval solution), sections were blocked with 3% hydrogen peroxide for 15 min. The primary antibodies for CD3 (18871, Santa Cruz, CA, USA), CD4 (13573, Santa Cruz), CD8 (18913, Santa Cruz), and TCRβ (H57-597, Novus, CO, USA) (1:200) were incubated overnight at 4  °C, respectively. Then, sections were incubated with a rabbit anti-mouse enhanced polymer antibody (pv-9000, Biosharp, Hefei, China), and reactions were developed in diaminobenzidine and nuclei counterstained with haematoxylin. Images were captured with a MoticAE 2000 microscope and quantified using Image J.

### 2.8. Flow Cytometry

The thymus and spleen were dissected and smashed through a 70-µm strainer (Thermo Fisher Scientific, Waltham, MA, USA). Mouse peripheral blood was obtained by a retro-orbital puncture. RBC lysis buffer (Qiagen) was used to lyse the red blood cells, and the cells were suspended in FACS buffer (PBS containing 0.1% BSA). Organoid-derived T cells were harvested by adding hyaluronidase (300 μg/mL, Sigma-Aldrich, St. Louis, MO, USA) to each well and were placed at 37 °C with a 5% CO_2_ incubator for hydrogel degradation overnight. The distribution of T cell subsets in the thymus or organoid was studied with four-color staining performed on freshly isolated thymocytes using the following antibodies: CD3 FITC (553061), CD4 APC (553051), CD8 PE (553032), and TCRβ Percp cy5.5 (H57-597) (all from BD Pharmingen, San Diego, CA, USA). T lymphocytes in the spleen and peripheral blood samples were analyzed using the following antibodies: CD3 APC (553066), CD4 FITC (553650), CD8 BV650 (563234), CD44 PE (553134) (all from BD Pharmingen), and CD62L Percp cy5.5 (104432) (Biolegend). All flow cytometry data were collected on a flow cytometer (CytoFLEX Beckman, CA, USA) and analyzed using FlowJo 10.5 software.

### 2.9. Cell Cycle Analysis

iTECs were collected and fixed with 70% precooled alcohol and then stored overnight at 4 °C. PI/RNase Staining Buffer (BD Pharmingen, San Diego, CA, USA) was used to dye the cells for 30 min in the dark, and cell lumps were filtered out through a 300-screen mesh, and then detected by flow cytometry (BD Accuri C6, CA, USA) and analyzed using Modfit LT 5.0 software (Verity Software House).

### 2.10. Apoptosis Detection

Apoptotic cells were analyzed by flow cytometry with an Annexin V-FITC/PI Apoptosis Detection Kit (BD Pharmingen, San Diego, CA, USA). Briefly, the thymocytes and different iTECs (10:1) were mixed and cultured for 24 h. The thymocytes were then resuspended with a binding buffer and incubated with Annexin V-FITC/PI and analyzed by BD Accuri C6.

### 2.11. Proliferation Assay

The thymocytes were stained with 2.5 μM CFSE (Sigma-Aldrich, St. Louis, MO, USA) for 20 min before the label was terminated with cold complete DMEM/F12. The labeled thymocytes were added to the upper chamber of an insert (8-μm pore diameter), and iTECs were planted in the lower chamber. After 2 days of incubation in the co-culture system, the thymocytes were detected by BD Accuri C6, and the cell proliferation rates were then analyzed using Modfit LT 5.0 software (Verity Software House).

### 2.12. RNA Isolation, Reverse Transcription and Quantitative Real-Time PCR

The total RNA was extracted from the cells by an RNA-Quick Purification Kit. Then, cDNA was synthesized using an M-MuLV First Strand cDNA Synthesis Kit, and real-time PCR was performed using SYBR Green PCR Mix. All reagents were from Sangon (Shanghai, China). The relative quantification of the genes was performed with the 2^−ΔΔCt^ method, and gene expression was normalized to *GAPDH*. The primers used in this study are listed in Appendix A.

### 2.13. TREC Quantification

Total spleen genomic DNA was extracted using the Mammalian genomic DNA extraction kit (Beyotime, Shanghai, China). The quantification of *TREC*s was performed by RT-qPCR using *TREC*, 5′-CATTGCCTTTGAACCAAGCTG-3′, 5′-TTATGCACAGGGTGCAGGTG-3′.

### 2.14. Cell Self-Renewal, Proliferation, and Viability Assays

The anchorage-independent growth of iTECs was estimated by a soft-agar colony formation assay as described. A total of 1000 cells were mixed in the 0.35% top agar and plated onto 0.5% basal agar, and incubated for 9 days at 37 °C. Pictures were taken with the inverted microscope (400×, Axio Observer. A1), and cell colonies larger than 0.1 mm were counted.

MTT and crystal violet assays were carried out to assess long-term cell survival. Briefly, 20 μL of MTT solution (5 mg/mL) was added to each well and incubated for 4 h, and MTT formazan was dissolved in 150 μL dimethyl sulfoxide (DMSO). For the crystal violet assay, 330 μL crystal violet dye was added into each well of the 24-well plate for 30 min, washed with water, and the plate was scanned at 570 nm in a microplate reader (EnSpire).

### 2.15. Migration Assays

Different iTECs were seeded into the inserts (8 µm) at a density of 2 × 10^4^/insert and cultured in the complete media to allow for cell migration. The migrated cells on the bottom surface were fixed with 4% PFA and stained with crystal violet for 15 min. The non-invaded cells were removed with cotton swabs. The number of stained cells from five random fields in each group was counted under an inverted microscope (Axio Observer. A1).

### 2.16. Statistical Analysis

All results are expressed as the mean ± SEM if not stated otherwise. All statistical analyses were calculated by performing the unpaired Student’s *t*-test, one-way ANOVA test with Tukey post-test, or two-way ANOVA analysis with Tukey post-test using Prism 9 (GraphPad Software). *p*-values < 0.05 were considered significant.

## 3. Results

### 3.1. Tβ15 Influences the Maturation of Thymus by Acting on TECs

The most variable thymus-related factors in acute aging mice were prothymosin alpha (PTMα) and thymosin beta 15 (Tβ15). We further overexpressed and silenced the corresponding thymosin, using immunohistochemistry and immunofluorescence to confirm the knockdown and overexpression of target genes (Figure 1C,D), and found no effects of PTMα on the thymic structure (Appendix A). In contrast, Tβ15 knockdown caused a marked reduction in the thymus sizes to 15.98% of the negative control group (*p* < 0.05), whereas high-level Tβ15 slightly inhibited the reduction in the thymus volume (Figure 1A). 

We then evaluated the change in the thymic architecture using H&E staining (Figure 1B). The area of the cortex in the Tβ15 OX thymus was significantly larger than that in the Tβ15 siRNA group, with thymocytes tightly arranged, and medulla shrank and became thin when compared with that of the controls, indicating a decreased thymic maturation. In contrast, in the Tβ15-knockdown thymus, the medulla expanded, but the number of mature thymocytes was significantly reduced (Figure 1B, *p* < 0.05). Furthermore, the cortex became thinner, and the thymocytes were sparse (*p* < 0.01).

To verify this event, we also varied the levels of Tβ15 in the thymic epithelial compartment in 3D organoid cultures to mimic thymopoiesis (Appendix A). Thymocytes were co-cultured for 3 weeks with Tβ15 OX iTECs; the organoid size was only half of the control group’s (*p* < 0.01), while their organoid forming efficiency was maintained at the original level (Appendix A). When co-cultured with Tβ15-knockdown iTECs, the size of the thymic organoids on day 7 increased by 1.81-fold compared to that of the controls. We also examined the effect of Tβ15 on apoptosis and the proliferation of thymocytes in vitro, and similar results were obtained.

Next, we engrafted constructed thymic organoids onto the axille of female BALB/c nude mice (Appendix A). The results showed that thymic organoids, when supported by Tβ15 sh iTECs, could significantly support the proliferation of thymocytes at 7 days, with the cell area ratio as high as 67.98% (Figure 1E–F, *p <* 0.01; Appendix A). Along with the maturation of organoids, there was a reduction in the HA-enriched area and growth in the formation of the iTEC and thymocyte complex within the organoids (Figure 1G–H, and Appendix A). These results indicate that high levels of Tβ15 maintain the low development level of the thymus, and Tβ15 knockdown causes a disturbed thymic architecture and accelerated maturation of thymocytes. We also prove that the effect of Tβ15 on the thymus structure and function is mainly due to its effect on TEC.

### 3.2. Tβ15 Inhibits the Development of mTECs and Suppresses the Formation of Reticulations in TECs

Our previous study showed that the distribution of the thymic cortex and medulla was affected by the level of Tβ15. We, therefore, used the TEC-associated markers CK5 and CK8 to study the differentiation and morphological changes in TECs in the thymus. When Tβ15 was at normal levels, three types of TECs were uniformly distributed in the thymus. However, a large increase in CK5 and CK8 positive cells could be observed in the medulla of the thymus or the middle of the organoids after Tβ15 knockdown (Figure 2C,F,G). In the Tβ15 overexpression thymus, the severe atrophy of the CK5^+^CK8^+^ regions was observed in the medulla (Figure 2A); instead, the number and Feret’s diameter (shown as Feret in Figure 2D) of mTECs was increased significantly, with 50–100 μm diameter cells representing more than half in the Tβ15 siRNA thymus (Figure 2B,D). We further found that Tβ15 overexpression reduced the proliferative ability of iTECs in vitro, which was consistent with a more immature medulla (Appendix A). Interestingly, the thymocytes showed substantial proliferation in the organoids after 2 weeks, together with CK5- and CK8-double positive TECs (Figure 2G–H).

We next examined whether the phenotype of thymic epithelia was affected by Tβ15. At week 1, the morphology of TECs was changed from spherical to reticular, indicating that it was differentiated into reticulations in all the organoids, and almost all TECs were a CK8^+^CK5^+^ phenotype (Figure 2E–F). Notably, the TECs were intertwined to form a denser mesh after Tβ15 knockdown (*p <* 0.01). After Tβ15 knockdown, several smaller CK5^+^CK8^+^ islands were formed that more closely resembled the structure of the thymus (Figure 2H, *p* < 0.01). We also found a few thymic nurse cells, full of thymocytes in the Tβ15 sh group but a lack of thymocytes in the Tβ15 OX group (Figure 2G), leading to a decline in TECs and in its support to the differentiation and maturation of thymocyte. This was consistent with the observations in organ culture systems, suggesting that Tβ15 knockdown might contribute to the ability of TECs to support thymocyte differentiation (Figure 2A–D). Collectively, these results suggest the potential role of Tβ15 in the regulation of mTEC maturation.

### 3.3. Tβ15 Delays the Development of CD4SP Thymocytes in Thymus

To assess whether Tβ15 could affect thymopoiesis, we analyzed the developmental stages of thymocytes based on the expressions of CD3, CD4, CD8, and TCRβ. Consistent with the imperfect medulla with high Tβ15 expression, we found that the thymus showed fewer CD3, CD4, CD8, and TCRβ expression in the thymocytes after 7 days of treatment (Figure 3A). Conversely, CD3, CD4, CD8, and TCRβ expression were significantly increased upon Tβ15 knockdown relative to that in the controls. Additionally, thymocytes with high TCRβ levels were found primarily at the subcapsular of the thymus (Appendix A). The organoid transplantation assay showed excessive levels of Tβ15 and restricted the differentiation and development of lymphocytes. At week 1 of organoid transplantation, the expression of CD3, CD4, and TCRβ in the organoid decreased after Tβ15 overexpression, but it had a slight effect on CD4 and CD8 expression (Figure 3B). However, Tβ15 knockdown distinctly increased CD3 and TCRβ expression, which was mainly distributed between the reticulations interleaved with the gel but did not affect the expression of CD4 and CD8. After two weeks, the expression of CD4 and CD8 in the Tβ15-OX group had still not increased, but the expression of CD3 and TCRβ significantly increased (Figure 3C). In comparison, Tβ15 knockdown increased the expression of CD4 and CD8. These findings suggest that the overexpression of Tβ15 blocks the maturation of thymocytes, whereas Tβ15-knockdown promotes thymus maturation.

Detailed analysis of the different stages of thymocyte revealed that the immature single-positive CD8^+^ (ISP8, CD3^−^TCRβ^−^CD4^−^CD8^+^) and double-positive (DP, CD3^−^TCRβ^−^CD4^+^CD8^+^) populations had already presented at week 1, but the double-negative (DN, CD3^−^TCRβ^−^CD4^−^CD8^−^) (13%–16%) cells remained constant throughout the culture (Figure 3D). These results indicated that the distribution of T cell subpopulations in the DN stage was not influenced by the Tβ15-mediated TECs developmental arrest. We next gated the CD3^+^TCRβ^+^ population to dissect the dynamics of the DP population selection process and found that the proportion of CD3^+^TCRβ^+^ cells slightly decreased after Tβ15 overexpression. However, Tβ15 knockdown increased the proportion of these cells from 13.0% to 24.0% (*p* < 0.01). The frequency of CD4SP (CD3^+^TCRβ^+^CD4^+^CD8^−^) production after Tβ15 overexpression was much lower than that in the controls (44.0% vs. 58.0%, *p* < 0.001). In contrast, the proportion of CD4SP cells increased by approximately 1.15-fold after Tβ15 knockdown (*p* < 0.05). It was also observed that the proportion of CDSP8 cells remained at approximately 13%–18% in all the groups. These results suggest that the conditional overexpression of Tβ15 blocks the differentiation and maturation of CD4SP thymocytes, and Tβ15 knockdown favors the lineage commitment to CD4SP over CD8SP.

Unlike the organ culture system, Tβ15 primarily affected the earliest stages of DN to DP development in thymic organoids (Appendix A), which might be related to the age of the mice. At week 2, there was a trend towards an increased frequency of DN after Tβ15 overexpression, and at week 3, the frequency of DN was 52.0%, which was approximately 1.28-fold higher than that of the controls (*p* < 0.001). Conversely, Tβ15 knockdown consistently decreased the frequency of DN from 53.9% (*p* < 0.05) at week 2 to 48.6% (*p* < 0.001) at week 3. Our results show an early block in DN cell development after Tβ15 overexpression and an acceleration toward more mature DN cell subsets after Tβ15 knockdown.

Taken together, these findings support that Tβ15 regulates the fated determination of thymocytes by affecting the directional differentiation of TECs.

### 3.4. Tβ15 Level in iTECs Influences Thymic Output and Activation of T Cells

In addition to the changes in the thymic structure already mentioned, we also observed that the thymus with low levels of Tβ15 directed the thymocytes to migrate toward the periphery (Figure 4A). The process of thymocytes exported from the thymus is also affected by cell differentiation, the maturation state, and sphingosine-1-phosphate receptor 1 (S1P1). The elevated expression of adhesion molecules, chemokines, and Spl in the Tβ15 siRNA-treated thymus also indicated an increased thymocyte migration following Tβ15 knockdown (Figure 4B). To further characterize the thymic function in thymic organoids transplanted mice, we quantified the T-cell receptor excision circles (TRECs) in the spleens of nude mice. TRECs is a maturation marker to assess thymic output. Our results showed a drastic increase in TRECs levels in the spleen after Tβ15 knockdown (week 1: *p* < 0.01, week 2: *p* < 0.01) compared to that in the controls and was maintained in mice with Tβ15 OX iTECs (Figure 4C). However, this phenomenon alone failed to explain its specific impact on the thymic output.

Functionally mature naïve T cells (defined herein as CD3^+^CD62L^+^CD44^neg/low^, T_N_) are excreted from the thymus and directed to secondary lymphoid organs, supported by the thymic output [22]. We assessed the activation of both CD4^+^ and CD8^+^ T cells in the peripheral blood and spleen using the CD62L and CD44 markers. Although Tβ15 had no effect on the frequency of splenic CD4^+^ T cells, the frequency of naïve CD8^+^ T cells was reduced, and memory T cells expanded upon the overexpression of Tβ15 (Figure 4D). Notably, the activation of CD4^+^ T cells was impaired after 2 weeks, and the frequency of effector memory (T_EM_, CD62L^−^CD44^hi^) and the acute/activation effector (T_AE_, CD62L^−^CD44^low^) in the T cells decreased by 2.71% and 5.06%, respectively (Figure 4E). Although mice with Tβ15-knockdown iTECs showed a significant increase in splenic TRECs levels, the frequency of CD8^+^ T cells decreased, and T_AE_ decreased from 68.86% to 43.86% as earlier.

In the peripheral blood, the frequency of CD4^+^ T cells increased by 0.72%, and that of CD8^+^ T cells decreased by 3.79% at week 1 after Tβ15 overexpression compared with that in the controls (Figure 4F). The results were the opposite for Tβ15 knockdown. In addition, the frequency in the memory compartment decreased in both cell subsets, suggesting that Tβ15 resulted in suboptimal functions of activation. After two weeks, the frequency of the CD4^+^ T cells in the peripheral blood of the mice with iTECs highly expressing Tβ15 increased to 9.63% (Figure 4G). Tβ15 knockdown and iTECs decreased by a CD8^+^ frequency, allowing immature thymocytes to enter the peripheral blood. Overall, these findings suggest that in the mice implanted with high-level Tβ15 iTECs, the main feature of the immune function is low thymus output, which contributes to the maturation and activation of peripheral T cells, whereas Tβ15 knockdown accelerates the immature thymocyte output after 14 days of transplantation.

### 3.5. Tβ15 Regulates Thymocyte Fate via Inhibiting the Directed Reticular Differentiation of TECs

As a class of G-actin-binding proteins, Tβ15 can alter the conformational and structural dynamics of actin. The epithelial protrusion, directional movement, and spatial distribution of epithelial cells are often driven by actin polymerization [23,24,25]. Therefore, the impact of Tβ15 on cell motility in iTECs was also assessed using the Transwell assay. The results showed the significantly enhanced migration of Tβ15 sh iTECs (Figure 5A); however, in Tβ15 OX iTECs, cell migration did not change significantly.

To gain insight into the potential role of Tβ15 genes in the differentiation of TECs, we comprehensively examined the levels of key factors in these processes (Figure 5B). RT-qPCR analysis indicated that mRNA levels of tumor necrosis factor-alpha (TNF-α, *p* < 0.01), interleukin-1beta (IL-1β, *p* < 0.001), and semaphorin-3A (Sema-3A, *p* < 0.01) were higher in Tβ15 OX iTECs than that in the controls. Comparatively, the levels of these factors in Tβ15 sh iTECs were significantly lower than those in the controls. Compared with the controls, Tβ15 sh iTECs showed an increased expression of interleukin-7 (IL-7, *p* < 0.05), leptin (*p* < 0.05), adrenocorticotropin hormone (ACTH, ns), fibroblast growth factor receptor-2IIIb (FGFR2IIIb, *p* < 0.05), forkhead-box n1 (Foxn1, *p* < 0.01), paired box gene 9 (Pax9, *p* < 0.01), delta-like 4 (Dll4, *p* < 0.05), and chemokine ligand chemokine 25 (Ccl25, *p* < 0.05), but their expressions were lower or similar in Tβ15 OX iTECs.

Tβ15 may further affect the differentiation and development of T cells by regulating the related factors secreted by TECs. In the positive selection, T-cell differentiation occurs through the T-cell receptor (TCR)-major histocompatibility complex (MHC) interactions between the T cells and TECs [26,27,28]. As shown in Figure 5D,E, the expression of L1 cell adhesion molecules (L1cam, *p* < 0.01) and histocompatibility 2-K1 (H2-K1, *p* < 0.001) decreased in Tβ15 OX thymus, but their expressions of Tβ15 sh-thymus significantly increased. In addition, although beta2-microglobulin (β2M, *p* < 0.001) increased in Tβ15 OX iTECs, the expression in the thymus was all at relatively low levels. Consistent with Tβ15 regulating TCR-induced apoptosis of the thymocytes (Appendix A), Tβ15 sh-thymus increased the expression of GTPase immune-associated protein 3 (Gimap3) and decreased the levels of GTPase immune-associated protein 4 (Gimap4) to reduce the apoptosis of immature DP or other thymocytes. Autoimmune regulators (Aire) and forebrain embryonic zinc fingerlike protein 2 (Fezf2) are unique promoters of the expression of tissue-specific antigens in mTECs [29,30]. The results showed that Aire (*p* < 0.001) and Fezf2 (*p* < 0.05) levels were reduced in the Tβ15 OX thymus compared to that in the controls, exactly the opposite of Tβ15 sh-thymus. Alternatively, although Aire (*p* < 0.001) and Fezf2 (*p* < 0.001) were expressed at rather low levels in Tβ15 sh iTECs, their expressions were not affected by Tβ15 overexpression.

In summary, our study demonstrates how the high expression of Tβ15 inhibites the network formation of mTECs and decreases the development and maturation of CD4SP thymocytes. Furthermore, the conditional knockdown of Tβ15 in iTECs accelerates the maturation of aTECs and mTECs along with CD4SP lineage commitment, causing an excessive thymic output but inconsistent T-cell activation status in the peripheral blood and spleen. All these results prove that Tβ15 regulates the spatial distribution and directed the development of TECs, while further regulating the mutual crosstalk between TECs and thymocytes, thereby affecting the differentiation fate of the latter. 

## 4. Discussion

In this study, we found that the thymus displayed an altered distribution in the cortex and medulla structure following the specific knockdown of Tβ15. Further research showed that Tβ15 critically regulated thymic development and maturation in a TEC-related manner. Tβ15 regulated the directional development and spatial distribution of TECs, particularly in mTECs. Mechanistically, Tβ15 regulated the thymic T cells development, negative/positive selection, and CD4SP lineage commitment by blocking the formation of mesh structures and directing the differentiation of TECs.

Tβ15 is an important peptide hormone secreted by subcapsular TECs; however, the importance of Tβ15 in TECs and its functional consequences on T cell development and tolerance have not been sufficiently investigated. TECs originate from epithelial progenitor cells (EPC) under the thymic capsule, and their spatial location determines their differentiation fate. EPC can differentiate into subcapsular, cortical astroid, and medullary reticular TECs. Among them, mTECs are structurally and functionally similar to aTECs. These two cells have multi-branched neurites, which are rich in the cytoplasm, and underlie the function of thymus nurse cells [31,32]. In the thymic organ culture system, the Tβ15 OX thymus resulted in a significant reduction in mTECs, whereas the numbers of mTECs and aTECs were increased in the Tβ15-knockdown thymus, especially Aire^+^ mTECs, protruding in a stellate reticulum manner. Actually, the anterior membrane of the epithelial progenitors of subcapsular TECs forms plate-like liposome neurites that drive the cells toward the medulla and reticulating differentiation. Therefore, we hypothesized that Tβ15 might block cTEC-like precursors that differentiated into mTECs.

The disorganized cortex-medulla structure and function, especially the imbalance of TEC distribution and function, can trigger an imbalance in thymocyte differentiation. The immature profile of the medulla is also reflected in the later stages of thymocyte development. We observed a decrease in the frequency of CD4SP thymocytes in the Tβ15 OX thymus. However, after two weeks, mice with high levels of Tβ15 in iTECs showed a significant increase in CD3^+^ and TCRβ^+^ cells in the organoids. This is possible because the differentiation of thymocytes was blocked, and T cell maturity became prominent at the DN and DP stages. In view of the above, we found that a high expression of Tβ15 did not appear to affect the exported number of T cells to the periphery, which favors the memory phenotype in T cell populations. 

In contrast, although mice with Tβ15 knockdown iTECs sustained the immune characteristics of a high thymic output by the upregulation of S1p1, Spl, and Icam-1, they still showed signs of slower peripheral T lymphocyte activity. At the earliest stages of organoid transplantation, we found that the proportion of the memory phenotype T cells decreased in both cell subsets, indicating that these mice had a poor early activation of T cells after Tβ15 knockdown. Furthermore, although it is generally believed that, unlike the endogenous thymus, thymic organoids are not continuously implanted with T-lymphocyte precursors. Without continuous progenitor cell implantation, the thymus independently maintains T lymphocylogenesis [33,34,35]. In our results, the thymocyte number of organoids in vivo far exceeded that of in vitro. This indicated that thymic organoids might also actively recruit bone marrow-derived hematopoietic progenitors in vivo. In conclusion, the low expression of Tβ15 improves the export of T cells to the periphery but in an incomplete maturation cell state, leading to decreased thymic immune function.

In this study, Tβ15 knockdown caused iTECs to display protrusions of the CK5-rich plasma membrane that interconnected with the adjacent iTECs to form 3D network structures (Appendix A). However, the formation of protrusions and the reticular differentiation of TECs are regulated by the dynamic assembly of actin. The dynamic balance of globular actin monomers (G-action) and polymerized filamentous actin (F-action) tightly regulates the process of assembly and the depolymerization of actin [36]. As the strongest actin-binding protein, Tβ15 was specifically highly expressed by the subcapsular TECs and mTECs in the thymus. We speculate that similar to Tβ4, Tβ15 also plays an important role in the assembly of G-actin in the TECs, depolymerization of F-actin, actin content in the cytoskeleton, formation of cell processes, and even the whole process of the movement and differentiation of TECs [37]. We hypothesized that Tβ15 affect the assembling of the actin filaments in the cytoskeletons of mTECs by binding to G-actin and then affect the spatial distribution of TECs and the reticular differentiation of mTEC, finally affecting the differentiation and output of thymocytes (See Figure 6).

The development, differentiation, and function of TECs are related to many factors, such as the hematopoietic factor IL-7 and pro-inflammatory factors (IL-1α/β, TNF-α). FGFR2IIIb and *leptin* regulate the reorganization of synaptic F-actin [38,39,40,41], and our results indicated how both of them were decreased in Tβ15-overexpressing iTECs, in which leptin could also affect the adhesion of cells to extracellular matrices [42,43]. Simultaneously, the low expression of *Foxn1* and its downstream gene Pax9 may hinder the differentiation of TECs. These results further supported the hypothesis that high levels of Tβ15 impaired the development of medullary compartments. We also found that ACTH and IL-1β expressions were regulated by Tβ15, but because of their immunomodulatory effects on Treg cells and B cells, functional studies of these cells are needed to draw more definitive conclusions [44,45,46]. Sema 3A is a soluble protein involved in cell adhesion and migration and can also reduce the number of F-actin-enriched protrusions [47,48,49]. Sema-3A was expressed in high levels in Tβ15 OX iTECs. The significant downregulation of the mRNA levels of IL-7, Dll4, and Ccl25 are the key factors supporting thymic T cell development and was also observed in Tβ15 OX iTECs. Our results further demonstrate that Tβ15 affects the development of TECs by the downregulation of key factors in TECs, thus leading to developmental defects in T lymphocytes.

TECs support T cell development and control and TCR repertoire selection through the secretion of different types of factors, especially Aire. The maturation of mTECs results in the upregulation of Aire, which is necessary for negative selection [50]. Interestingly, on a per-cell basis, the levels of Aire and Fezf2 were contrasted between the Tβ15-knockdown thymus and iTECs. This implies that Tβ15 did not directly regulate Aire and Fezf2 expressions. Unlike negative selection, positive selection depends greatly on cTECs. We also identified five genes associated with the positive selection of DP thymocytes, including intrinsic genes, such as Gimap3 and Gimap4 [51,52], which are thought to be expressed by thymocytes. Extrinsic genes, such as H2-K1 and β2m, are components of MHC I molecules, and L1cam plays an important role in the class MHC II-mediated peptide presentation in TECs [53,54,55]. These extrinsic genes provide peptide-MHC ligands for thymocyte development and are bidirectionally regulated by Tβ15. In addition, L1cam provided survival information to thymocytes (especially CD4SP cells); however, the L1cam levels in these cells were relatively low, indicating that the maturation of CD4SP might be impaired by the downregulation of L1cam in the thymus. Thymocytes co-cultured with Tβ15-knockdown iTECs largely inhibited TCR-induced thymocyte apoptosis. In addition, Gimap3 and Gimap4 expression did not obviously change in the Tβ15 OX thymus, although they are both antagonistic to thymocyte apoptosis. In conclusion, this allows us to provide an additional mechanism of immune dysregulation, where Tβ15 leads to defects in positive selection from DP to SP T cells by inhibiting the production of aTECs.

Overall, our findings strongly suggest that Tβ15 participates in governing the development, motility, and reticular differentiation of TECs, regulating the positive selection of thymocytes and the homeostasis of peripheral T cells. In the normal level of Tβ15, TECs ensure the normal structure and function of the thymus, keeping the immune balance of T cells in the body. High-level Tβ15 leads to the incomplete maturation of mTECs, loose mesh structures, the defective development of CD4SP thymocytes, and interferes with the homeostasis of peripheral T lymphocytes. However, low levels of Tβ15 in TECs shift TECs towards the reticular differentiation to aTECs and mTECs, limiting the commitment of the CD4SP lineage and disrupting thymic output, making a large number of immature T cells flood the peripheral blood. Our present results further extend our understanding of the biological functions of Tβ15 and provide new insights into its mechanisms for controlling the directed and spatial development of TECs and positive selection during T-cell development. It is of great significance to further explain the mechanism of thymus development and degeneration in different physiological and pathological conditions.

## Figures and Tables

**Figure 1 cells-11-03679-f001:**
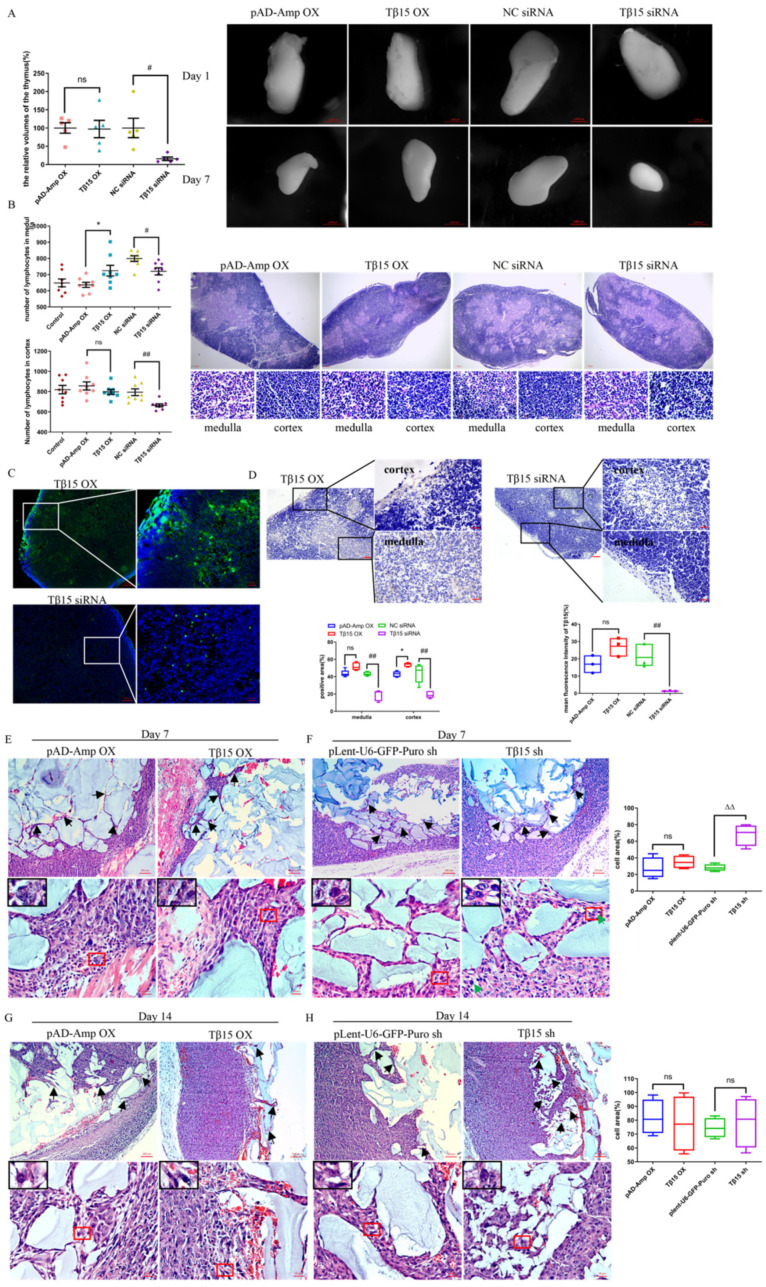
Tβ15 results in impaired thymic development. (**A**) Representative images of the thymus tissues transduced with Tβ15 siRNA or Tβ15 OX for 7 days. Data represents the relative volumes of the thymus at day 7 (%). Scale bar = 1000 μm. (**B**) Representative thymus histomorphology sections in different groups. Scale bar = 100 μm (upper) and 10 μm (lower). Data represents cell numbers in the cortex and medulla. (**C**,**D**) Representative images of Tβ15 knockdown or overexpression in thymus were determined by immunofluorescence (**C**) and immunohistochemistry staining (**D**). Data represents efficiencies of Tβ15 knockdown or overexpression in thymus. Scale bar = 100 μm (left) and 10 μm (right). (**E**–**H**) The H&E staining of thymic organoids from the iTEC-specific Tβ15 altered *(n* = 3) and littermate control mice (*n* = 3) at week 1 (**E**,**F**) and 2 (**G**,**H**) of transplantation is shown. Black arrowheads show reticular structures and green arrowheads show reticular cells. Scale bar = 100 μm (upper) and 10 μm (lower). Data are expressed as the percentage of the cell proliferation area in the organoid to the entire organoid area (except for the skin). Each symbol represents individual data, all data are pooled from at least three independent experiments and presented as mean ± SEM. A one-way ordinary ANOVA test was performed, * *p* < 0.05, compared to pAD-Amp OX group. ^#^
*p* < 0.05, ^##^
*p* < 0.01 compared to NC siRNA group. ^△△^
*p* < 0.01, compared to plent-U6-GFP-Puro shRNA group, and ns = not statistically significant.

**Figure 2 cells-11-03679-f002:**
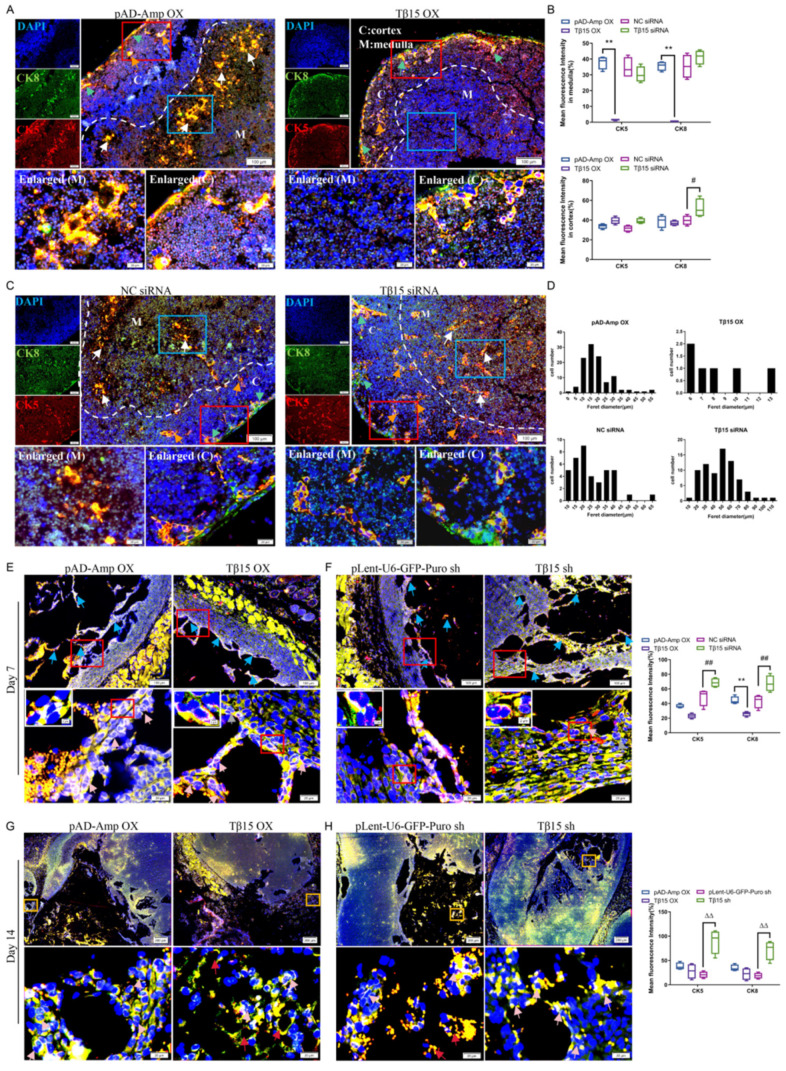
Tβ15 inhibits cTECs for differentiation into mTECs, and suppress the formation of a reticular structure. (**A**,**C**) Representative immunofluorescence analysis for CK5 (red) and CK8 (green) of the thymic tissue with Tβ15 siRNA or Tβ15 OX. Nuclei were stained with DAPI. White arrows show mTECs; green arrows show subcapsular TECs, and orange arrows show aTECs: the dashed line outlines of cortical and medullary thymic regions. Scale bar = 100 μm (upper) and 20 μm (lower). (**B**) The mean fluorescence intensity of CK5 and CK8 in the cortex and medulla. (**D**) Size distribution and particle diameters in mTECs. (**E**–**H**) Representative immunofluorescence staining of thymic organoids from the iTECs-specific Tβ15 altered (*n* = 3) and littermate control mice (*n* = 3) at week 1 (**E**,**F**) and 2 (**G**,**H**) of transplantation. Data are expressed as the mean fluorescence intensity of CK5 or CK8 in thymic organoids. Blue arrowheads show reticular structures, pink arrowheads show nurse cells, and red arrowheads show epithelial voids. In (**E**,**F**), Scale bar = 100 μm (upper) and 20 μm (lower). In (**G**) and (**H**), Scale bar = 200 μm (upper) and 20 μm (lower). Data are representative of at least two biological replicates in each group and yield essentially the same results, ** *p* < 0.01 compared to pAD-Amp OX group. ^#^ *p* < 0.05, ^##^
*p* < 0.01, compared to NC siRNA group. ^△△^
*p* < 0.01, compared to plent-U6-GFP-Puro shRNA group, and ns = not statistically significant.

**Figure 3 cells-11-03679-f003:**
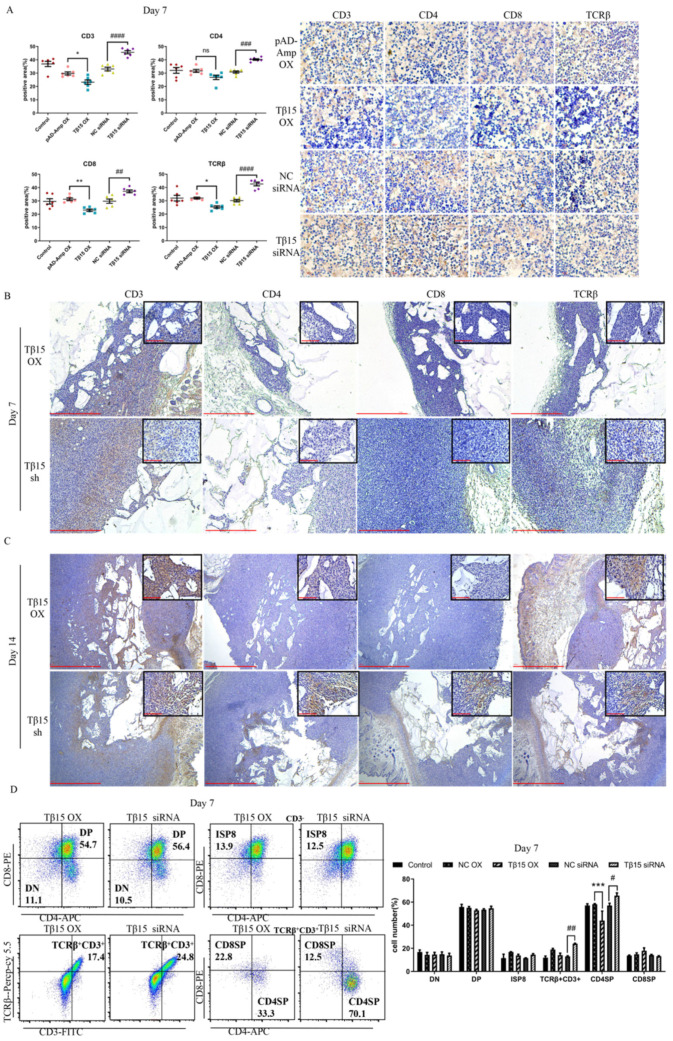
Tβ15 causes an impaired development of mature CDSP4 thymocytes. (**A**) Representative images of IHC staining in thymus tissues transduced with Tβ15 siRNA or Tβ15 OX for 7 days. Data are representative of the positive staining areas analyzed by Image J. Scale bar = 10 μm. (**B**,**C**) IHC staining of thymic organoids from the iTECs-specific Tβ15 altered (*n* = 3) and littermate control mice (*n* = 3) at week 1 (**B**) and 2 (**C**) of transplantation is shown. Scale bar = 1000 μm (main figure) and 100 μm (inset). (**D**) Comparison of T cell differentiation in thymus transduced with Tβ15 siRNA or Tβ15 OX for 7 days. Frequencies of DN cells (CD3^−^TCRβ^−^CD4^−^CD8^−^), immature single-positive CD8^+^ (ISP8) cells (CD3^−^TCRβ^−^CD4^−^CD8^+^), double-positive (DP) cells (CD3^−^TCRβ^−^CD4^+^CD8^+^), and TCRβ^+^CD3^+^ are shown as percentage of total cells. CD8SP cells (CD3^+^ TCRβ^+^CD4^−^CD8^+^) and CD4SP cells (CD3^+^TCRβ^+^CD4^+^CD8^−^) are shown as percentage of TCRβ^+^CD3^+^ cells in the thymus. All data are pooled from at least three independent experiments and present as mean ± SEM. A one-way ordinary ANOVA test was used in (**A**), two-way ANOVA test in (**D**): * *p* < 0.05, ** *p* < 0.01, *** *p* < 0.001 compared to pAD-Amp OX group. ^#^
*p* < 0.05, ^##^
*p* < 0.01, ^###^
*p* < 0.001, ^####^ *p* < 0.0001 compared to NC siRNA group. ns = not statistically significant.

**Figure 4 cells-11-03679-f004:**
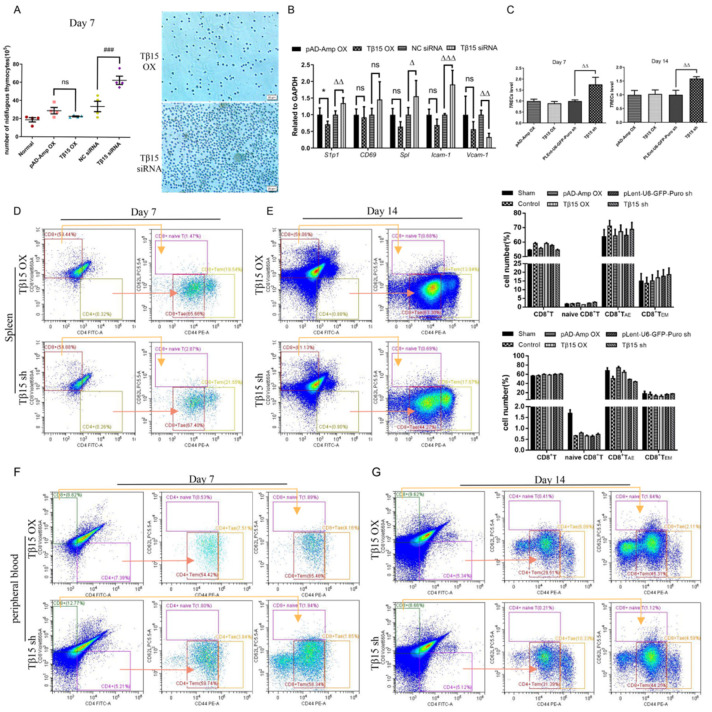
Differential expression of Tβ15 caused impaired thymic output in mice. (**A**) Representative images of thymocytes that migrated from thymus tissues transduced with Tβ15 siRNA or Tβ15 OX for 7 days. Data represents the number of thymocytes moving out of the thymus (10^5^). Scale bar = 20 μm. (**B**) The mRNA levels of S1p1, CD69, Spl, Icam-1, Vcam-1 in different groups. For each sample the mRNA level was normalized using the corresponding GAPDH mRNA level. (**C**) Changes in the expression of TRECs in the spleen of each group (*n* = 3) at week 1 and 2 of transplantation. (**D**–**G**) Comparison of T cell differentiation in spleen (**D**,**E**) and peripheral blood (**F**,**G**). Frequencies of CD4^+^ T cells (CD3^+^CD4^+^CD8^−^) and CD8^+^ T cells (CD3^+^CD4^−^CD8^+^) are shown as percentage of total cells. Frequencies of CD4^+^ naïve T cells (CD3^+^CD4^+^CD8^−^CD62L^+^CD44^neg/low^), CD4^+^ acute/activation effector memory (AE) T cells (CD3^+^CD4^+^CD8^−^CD62L^−^CD44^low^), and CD4^+^ effector memory (EM) T cells (CD3^+^CD4^+^CD8^−^CD62L^−^CD44^hi^) are shown as percentage of CD3^+^CD4^+^CD8^−^ cells. CD8^+^ naïve T cells, CD8^+^ acute/activation effector memory T cells, and CD8^+^ effector memory T cells are shown as percentage of CD3^+^CD4^−^CD8^+^ cells. In (**A**–**C**), data are representative of at least three independent experiments and used a one-way ordinary ANOVA test. * *p* < 0.05, compared to pAD-Amp OX group. ^###^
*p* < 0.001 compared to NC siRNA group. ^△^
*p* < 0.05, ^△△^
*p* < 0.01, ^△△△^
*p* < 0.001 compared to plent-U6-GFP-Puro shRNA group, and ns = not statistically significant. In D and E, data are representative of two independent experiments and yield essentially the same results.

**Figure 5 cells-11-03679-f005:**
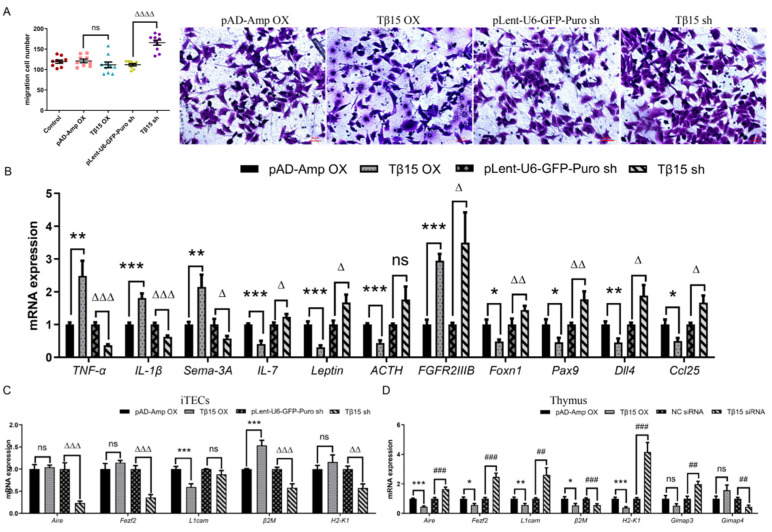
Tβ15 inhibits the formation of the network structure and directional development of TECs, hindering the development of thymocytes. (**A**) The effect of Tβ15 on migration in iTECs is assessed using the Transwell assay. Data represents the average number of migrated cells of three independent experiments. Scale bar = 100 μm. (**B**) The mRNA levels of TNF-α, IL-1β, Sema-3A, IL-7, leptin, ACTH, FGFR2IIIb, Foxn1, Pax9, Dll4, and Ccl25 in the iTECs of different groups. For each sample, the mRNA level was normalized using the corresponding GAPDH mRNA level. (**C**,**D**) The mRNA levels of Aire, Fezf2, L1cam, β2M, H2-K1, Gimap3, and Gimap4 in different groups. For each sample the mRNA level was normalized using the corresponding GAPDH mRNA level. Each symbol represents individual data; all data are pooled from three independent experiments and presented as mean ± SEM. A one-way ordinary ANOVA test was performed in (**B**), nonpaired Student’s t-test was used in (**D**,**F**). * *p* < 0.05, ** *p* < 0.01, *** *p* < 0.001 compared to pAD-Amp OX group. ^##^
*p* < 0.01, ^###^
*p* < 0.001 compared to NC siRNA group. ^△^
*p* < 0.05, ^△△^
*p* < 0.01, ^△△△^
*p* < 0.001, ^△△△△^
*p* < 0.0001 compared to plent-U6-GFP-Puro shRNA group, and ns = not statistically significant.

**Figure 6 cells-11-03679-f006:**
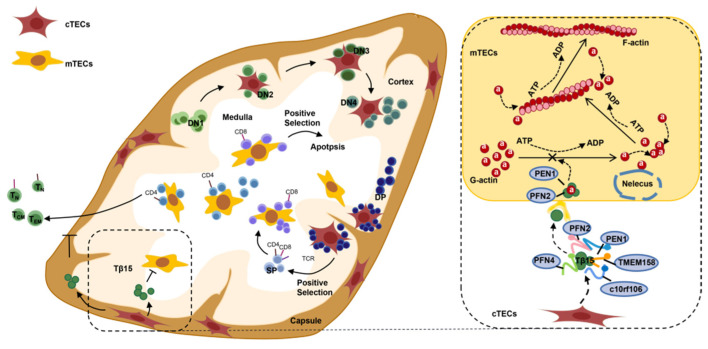
Schematic representation of Tβ15 regulating the fate determination of thymocytes by changing TEC development. In the cortex, thymocytes differentiate through CD4^−^CD8^−^ double-negative (DN) stages to the CD4^+^CD8^+^ double-positive (DP) stage and undergo positive selection; these events are mediated by cortical thymic epithelial cells (cTECs). Then, in the medulla, mTECs facilitate the elimination of self-reactive thymocytes to prevent autoimmunity. In this process, thymosin β15 (Tβ15) is secreted by subcapsular TECs by binding to free G-actin, which affects the cytoskeletal actin filaments of mTECs and, thereby, affecting thymocyte development.

## Data Availability

Publicly available datasets were analyzed in this study. This data can be found here: [https://pan.baidu.com/s/1hk2wvgHsbbWd6rYu5DujXA?pwd=65C1, accessed on 18 October 2022].

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
