# Peer review of "Thymosin Beta 15 Alters the Spatial Development of Thymic Epithelial Cells"

_cells, 2022, doi:10.3390/cells11223679_

Round 1

Reviewer 1 Report (Previous Reviewer 2)

The authors have done a good job addressing the initial concerns, and have added statistical rigour to their findings. 

Reviewer 2 Report (Previous Reviewer 3)

Authors have addressed all the comments. I have no further comments.

This manuscript is a resubmission of an earlier submission. The following is a list of the peer review reports and author responses from that submission.

Round 1

Reviewer 1 Report

Authors should address the following points:

1. The rationale for assessing thymosin beta -15 should be explained in more details.

2. Can the authors analyze any publicly available or their global transcriptomics data to show what are the expression pattern of other thymosin beta proteins?

3. Please provide statistical estimations for Figure 2. You may sample different fields for each biological sample.

4. How do authors explain different response of TEC and CD4SP to thymosin experresion?

5. How do authors explain the relative levels of IL-7, Leptin, ACTH, PAX9 in thymosin beta 15 over expression and knockdown samples?

6. Figure 6 should be remade and more explanatory.

Reviewer 2 Report

The manuscript by Xu et al. explores the role of thymosin beta15 (Tb15) in regulating thymic epithelial cell (TEC) function and thymocyte development.  Thymosins are secreted polypeptides with hormone-like functions that were initially characterized as potentially immune-modulatory. However, what these factors normally do and how they function remains unclear.  The authors take advantage of two model systems to examine the potential function of Tb15: 1) the use of thymus organ cultures transduced with adenoviruses to knockdown or overexpress Tb15; and, 2) used iTEC lines also transduced to knockdown or overexpress Tb15.  In some cases, the iTECs were aggregated with total thymocytes to form artificial thymus organoid cultures.  The overall approach is commendable but suffers from unclear descriptions of the work and could benefit from better quantitation of the findings.  Nevertheless, it is clear that modulating Tb15 expression does lead to changes in thymic size, TEC subset distribution and CD4+CD8- development.

Major Concerns:

1-    The results shown using the iTEC organoid system are highly variable and lack quantitation, making these findings unconvincing and detracting from the rest of the work making using of thymus organ cultures.  The results from the iTEC organoid work require statistical rigor and better ways to quantify the changes that are claimed to be taking place, as shown in Fig1 C-F; Fig2 B-D; and, Fig3 B-C.  

2-    What are “spilusive” thymocytes? As per Figure 4, which also has fairly unconvincing results in sections D-G.  Plus, what are “nidifugous” thymocytes, as per the y-axis in section A.  Overall, Figure 4 does not help to support the authors claims. 

3-    One of the claims made by the authors is that the work addresses thymic involution and/or premature senescence. However, there is no evidence given in the work to point to either mechanism, as no effort is made to use or address thymuses from aged mice or undergoing immune-senescence.  These claims should be removed or highly tempered. 

4-    The iTEC results shown in Figure 5 C-D, are intriguing and could be more valuable if these cells are also used to validate the approach, i.e., show that Tb15 expression is in fact lower in sh or si samples, and that it is overexpressed in the OX samples, perhaps show a Western blot and/or Elisa to show changes in Tb15 expression. 

Additionally, the manuscript needs editing to improve the syntax and usage, for instance there are many sentences that make no sense at all, eg: Of note the localization of TECs in thymocytes was clearly distinguished after Tβ15 knockdown (line 284-285), and In the action of environmental effects and extracellular adhesion proteins, the anterior membrane of epithelial progenitors of subcapsular TECs form plate-like liposome neurites that driving cells toward the medulla and reticulating differentiation (line 520-523).  The meaning of these and many other sentences are very elusive at best. 

Reviewer 3 Report

Xu et al., investigated the role of Tβ15 in TECs differentiation and thymic immunosenescence. This is an interesting study with using in vivo system. However, authors should further address the following comments to make this manuscript more interesting:

1.      Figure 1B: Tβ15 siRNA thymic size is not smaller compared to control, this difference is very much clear in Figure 1A. Fig 1A and 1B are from the same tissue?

2.      Line 270-271: refer actual figures. For figure 2B use white arrow and put colour coding of representative markers.

3.      Result 3: CD8 and CD4 are assessment of mature T cell population, apart from T cell immunogenic marker did you check any other epithelial marker?

4.      Result 4: as per given datasets, silencing Tβ15 do not affect thymocyte integrity (low count-Fig1A-B) only it induces their migration to secondary lymphoid organs (spleen). Any evidence of cytokines/chemokines alteration in this process?

5.      It is not clear why authors have not measured the intracellular levels of Tβ15 in thymocytes? Or any drug induced effect on Tβ15 levels? RNA data is not sufficient.

6.      It is also possible that without antigenic maturation of T cells, they may acquire immunosenescence. Authors comment.

7.      Line 57-60: please rephrase the sentences, it’s not clear.